# Meta-Adapters: Parameter Efficient Few-shot Fine-tuning through Meta-Learning

**Trapit Bansal**[*1]  **Salaheddin Alzubi**[1]  **Tong Wang**[2]  **Jay-Yoon Lee**[1]  **Andrew McCallum**[1]

[1]University of Massachusetts Amherst
[2]Microsoft Research, Montréal

**Abstract**   Consistent improvements in the representational capacity of large pre-trained transformers has made it increasingly viable to serve these models as shared priors that can be fine-tuned on a large number of downstream tasks. However, fine-tuning the entire model for every task of interest makes a copy of all the model parameters, rendering such scenarios highly impractical. Recently introduced Adapter methods propose a promising alternative, one where only a small number of additional parameters are introduced per task specifically for fine-tuning. However, Adapters often require large amounts of task-specific data for good performance and don't work well in data-scarce few-shot scenarios. In this paper, we approach parameter-efficient fine-tuning in few-shot settings from a meta-learning perspective. We introduce *Meta-Adapters*, which are small blocks of meta-learned adapter layers inserted in a pre-trained model that re-purpose a frozen pre-trained model into a parameter-efficient few-shot learner. Meta-Adapters perform competitively with state-of-the-art few-shot learning methods that require full fine-tuning, while only fine-tuning 0.6% of the parameters. We evaluate Meta-Adapters along with multiple transfer learning baselines on an evaluation suite of 17 classification tasks and find that they improve few-shot accuracy by a large margin over competitive parameter-efficient methods, while requiring significantly lesser parameters for fine-tuning. Moreover, when comparing few-shot prompting of GPT-3 against few-shot fine-tuning with Meta-Adapters, we find that Meta-Adapters perform competitively while working with pre-trained transformers that are many orders of magnitude (1590×) smaller in size than GPT-3.

## 1 Introduction

Pre-trained models in natural language processing (NLP) have consistently increased in size over time (Devlin et al., 2019; Raffel et al., 2019; Brown et al., 2020). These models are often used as initialization for transfer learning, where the initialized model is fine-tuned on a task of interest. However, when such pre-trained models are intended to be served for many downstream tasks at once, such as in a cloud-based machine learning (ML) service, then full fine-tuning necessitates keeping as many parameter copies as the number of tasks – rendering them extremely inefficient. An alternative to full fine-tuning is Adapters (Houlsby et al., 2019). Adapters add a small number of randomly initialized parameters to a pre-trained model such that fine-tuning only the Adapters, freezing the rest of the pre-trained model, still performs competitively with full fine-tuning.

In this paper, we consider the scenario where we want to deploy a shared model for a large number of tasks, in an online setting, such that models can be quickly adapted to target tasks without access to a lot of data. An example of such a setting is a cloud-based ML service which allows users to specialize models to their own NLP tasks with scarce training data. Adapters are particularly useful in such scenarios as they allow sharing a pre-trained model backbone across tasks. However, adapters are randomly initialized blocks of parameters which can perform poorly when the target

---

*Corresponding author. Work done while at UMass Amherst, Trapit's currrent affiliation is OpenAI.

task has few examples. Such scenarios pose a dual problem: one of enabling parameter efficient fine-tuning, and another of accurate few-shot learning.

Meta-learning (Schmidhuber, 1987; Bengio et al., 2003; Thrun and Pratt, 2012) is often employed to learn effective few-shot learning models, that can generalize to new unseen tasks with small amounts of labelled data by learning from a distribution of other related tasks. Within NLP, meta-learning models have been developed for few-shot learning on a diverse range of NLP tasks (Han et al., 2018; Brown et al., 2020; Bansal et al., 2020a). Of particular interest in this work are gradient-based methods (Finn et al., 2017) that learn a model initialization to enable few-shot learning with a few steps of gradient descent. By directly optimizing the training for few-shot fine-tuning, these methods help mitigate the train-test mismatch in few-shot learning and enable effective generalization to new few-shot tasks. However, existing applications of such meta-learning methods (Bansal et al., 2020a,b; Dou et al., 2019) don't leverage existing pre-trained models and fine-tune the entire model making them inefficient when applied to many tasks.

We thus develop a meta-learning model that enables accurate and parameter-efficient few-shot learning – utilizing a shared, *frozen* pre-trained model backbone that can rapidly adapt to downstream tasks with only a handful of additional parameters and labeled data per new task. Our approach re-purposes an existing pre-trained transformer model into an efficient few-shot learner by introducing *Meta-Adapters*, a small number of meta-learned parameters that modulate the pre-trained models activations to make them effective for few-shot learning. Our objective is to enable parameter efficient few-shot learning at inference time; the Meta-Adapters are *trained* to "prime" the regular adapter towards this objective on a wide variety of few-shot tasks resembling the target tasks (Section 3). Moreover, Meta-Adapters are more efficient to train than contemporary meta-learning models as they only train a subset of the full model. On a suite of 17 few-shot classification tasks, our results indicate that Meta-Adapters are better than randomly initialized adapters (Houlsby et al., 2019) for few-shot learning, are more accurate and efficient than multi-task fusion adapters (Pfeiffer et al., 2021), and perform competitively with previous state-of-the-art meta-learning methods that involve full fine-tuning (Bansal et al., 2020b), while only adding 0.6% model parameters per task (Figure 1). Comparing Meta-Adapters with the alternative few-shot prompting approach of GPT-3 (Brown et al., 2020), we find that fine-tuning with Meta-Adapters performs better on average than the largest GPT-3 model (175 Billion parameter DaVinci) while operating on pre-trained models that are order of magnitude smaller in parameter size (110 Million BERT-base).

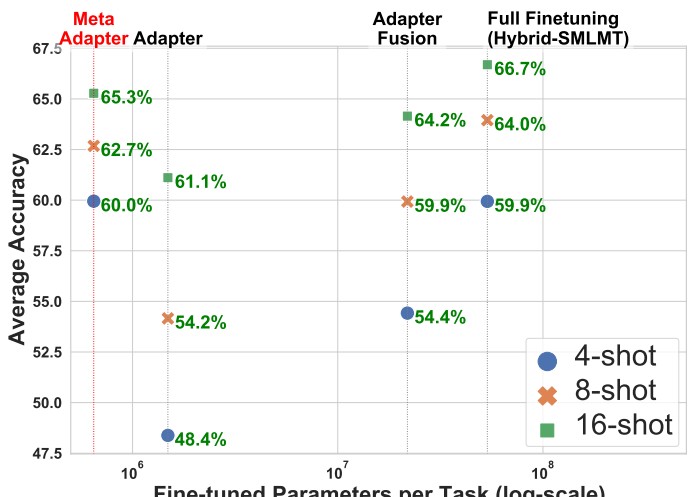

Figure 1: Comparison of overall average accuracy across 17 tasks vs the number of parameters fine-tuned per task (on a log scale). Meta-Adapters fine-tune only 0.6% of total model paramters per task, are more efficient and accurate than other adapter alternatives, and competitive with a meta-learning approach that requires full fine-tuning.

## 2 Background

Adapters (Houlsby et al., 2019) are blocks of feedforward layers, comprising of a downward projection followed by an upward projection, that are added between subsequent layers of a pre-

trained transformer model. Let $\theta$ denote the parameters of the transformer and $\phi$ the parameters of the adapters. Then given a target task $T$, with some data, $\mathcal{D}_T^{tr}$, and loss function, $\mathcal{L}_T(\cdot)$, adapters minimize the following objective using a gradient descent routine, termed as fine-tuning:

$$\min_{\phi} \mathcal{L}_T(\theta, \phi; \mathcal{D}_T^{tr}) \tag{1}$$

where adapters $\phi$ are often initialized randomly (Houlsby et al., 2019). Note that the size of $\phi \ll \theta$, leading to parameter savings when the same model parameters $\theta$ are re-used for many tasks $\{T\}$.

However, as $\phi$ are randomly initialized they may not perform well in the few-shot setting where $\mathcal{D}_T^{tr}$ is very small, for instance when there are only 4 examples per label. Moreover, the original pre-trained model is not optimized for few-shot learning and can lead to sub-optimal performance (Bansal et al., 2020b).

Alternatively, few-shot problems are often formulated as meta-learning problems. We refer the reader to Hospedales et al. (2020) for a comprehensive review. Our work builds on model agnostic meta-learning (MAML) (Finn et al., 2017) which, given a distribution over tasks, learns a model initialization for better few-shot learning with a few steps of gradient descent. This involves an inner loop of task-specific fine-tuning and an outer loop of optimizing the inner loop performance across tasks. Note that the inner loop corresponds directly to the inference method applied to any new task, that is, gradient-based fine-tuning. MAML-based methods have been explored in prior work for improving few-shot learning (Dou et al., 2019; Bansal et al., 2020b). However, these methods require fine-tuning the entire network at inference time and optimizing the entire model parameters at training time. This makes fine-tuning very inefficient when applied to many tasks at once and also doesn't leverage existing self-supervised models pre-trained on large amounts of unlabeled data.

## 3 Meta-Adapters

Our goal for parameter efficient learning is two-fold: (1) leverage and re-purpose existing pre-trained model into a better few-shot learner; (2) make fine-tuning parameter efficient by sharing the pre-trained model backbone and introducing only a fraction of parameter overhead for each new task.

We thus introduce Meta-Adapters, which are meta-learned adapter layers inserted between layers of a frozen pre-trained model to improve performance in few-shot learning. Meta-Adapters have the same architecture as feed-forward adapter layers (Houlsby et al., 2019) and differ in their placement in the model architecture, their training and usage. Whereas adapters are randomly initialized and fine-tuned per task, Meta-Adapters are trained parameters that are not fine-tuned on new tasks but instead modulate the activations of the pre-trained model in the forward and backward pass during fine-tuning to allow better few-shot learning. Figure 2 shows an overview of the approach.

Meta-Adapters operate in conjunction with regular adapters and are trained to enable parameter-efficient few-shot learning. In particular, consider a transformer model layer with adapters added after the two sets of feed-forward blocks, as shown in Fig.2. The Meta-Adapters layers sandwich the adapter layer from above and below, and consist of a two-layer feed-forward network with a downward projection bottleneck. The bottleneck dimension is typically small, a hyper-parameter $\leq 32$ in our experiments, that keeps

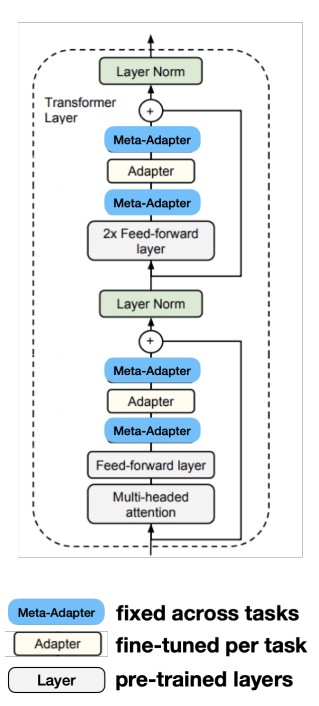

Figure 2: Meta-Adapters architecture.

the number of Meta-Adapters parameters manageable. During the Meta-Adapters training phase, it is optimized to improve the regular adapter fine-tuning with few-shot training task data. During inference, each few-shot target task is then solved by fine-tuning only the regular adapters, freezing the rest of the model to achieve parameter efficiency.

Denoting $\omega$ as the Meta-Adapter parameters, $\phi$ as the adapter parameters, and $\theta$ as the pre-trained transformer parameters, the objective for each individual task, T, remains similar to regular adapters:

$$\phi_T \leftarrow \arg\min_\phi \mathcal{L}_T(\theta, \phi, \omega; \mathcal{D}_T) \tag{2}$$

Note that $\omega$ is not fine-tuned for individual task T but it still modulates the activations in the forward pass as well as the backward pass. Thus, $\omega$ needs to be optimized to directly improve adapter fine-tuning with few-shot data, which leads to the following objective, where $\phi_T$ is obtained from the minimization in (2):

$$\min_\omega \mathbb{E}_T \left[ \mathcal{L}_T(\theta, \phi_T, \omega; \mathcal{D}_T) \right] \tag{3}$$

Computing these nested minimization to convergence will be computationally infeasible. We thus approximate these by few-steps of gradient descent. This can then be formulated as a meta-learning problem involving bi-level optimization, and is related to model agnostic meta-learning (MAML). We use the episodic framework (Vinyals et al., 2016; Finn et al., 2017) for solving the problem in equation (3), where each episode samples a few-shot task with a training data $\mathcal{D}^{tr}$ and validation data $\mathcal{D}^{val}$. $\mathcal{D}^{tr}$ is then used for the minimization in (2) and $\mathcal{D}^{val}$ is used for the minimization in (3). This leads to the following inner and outer loop updates for training the Meta-Adapters:

$$\text{Inner:} \quad \phi_T' \leftarrow \phi - \alpha \nabla_\phi \mathcal{L}_T(\theta, \phi, \omega, \mathcal{D}_T^{tr}) \qquad \textit{\# fine-tune Adapters} \tag{4}$$

$$\text{Outer:} \quad \omega \leftarrow \omega - \beta \nabla_\omega \mathbb{E}_{T \sim \mathcal{P}(\mathcal{T})} \left[ L_T(\theta, \omega, \phi_T', \mathcal{D}_T^{val}) \right] \qquad \textit{\# train Meta-Adapters} \tag{5}$$

$$\phi \leftarrow \theta - \beta \nabla_\phi \mathbb{E}_{T \sim \mathcal{P}(\mathcal{T})} \left[ L_T(\theta, \omega, \phi_T', \mathcal{D}_T^{val}) \right] \qquad \textit{\# train Adapters initialization}$$

$$\alpha \leftarrow \theta - \beta \nabla_\alpha \mathbb{E}_{T \sim \mathcal{P}(\mathcal{T})} \left[ L_T(\theta, \omega, \phi_T', \mathcal{D}_T^{val}) \right] \qquad \textit{\# train fine-tuning learning-rates}$$

The inner loop (4) is carried out for multiple steps of gradient descent. Through these steps, note that we also learn an initialization of the adapters $\phi$ in preparation for few-shot learning, in addition to training the Meta-Adapters $\omega$. Thus, there is no random initialization for adapters, nor the selection of hyper-parameters for the initialization (like scale), that needs to be set for each down-stream task. In addition, we also treat the inner loop learning rate $\alpha$, in (4), as a learnable parameter. We use different learning rates for each layer. The inner loop directly corresponds to the fine-tuning procedure on any new task, thus this removes the requirement to set a crucial hyper-parameter for each new task as the learned learning rates are re-used for fine-tuning on new tasks.

**Training Tasks:.** Meta-learning the Meta-Adapters (equation 4, 5) requires a distribution of tasks $\mathcal{P}(\mathcal{T})$, as is typical in meta-learning methods (Vinyals et al., 2016; Finn et al., 2017). Tasks are sampled from this distribution to learn models for few-shot learning. Ideally, this distribution of tasks should be large and diverse to enable learning of effective models that can generalize to new tasks. We follow prior work (Bansal et al., 2020b) and use a combination of supervised and unsupervised tasks to provide a diverse distribution of training tasks. The supervised tasks come from the set of GLUE tasks (Wang et al., 2018) that comprise of 8 diverse tasks requiring sentence-level understanding. In addition we use the cloze-style SMLMT tasks proposed in Bansal et al. (2020a). These are self-supervised, blank-filling tasks (Devlin et al., 2019), that are automatically created from unlabeled text and were shown to be a useful source of meta-training tasks for few-shot learning. In particular, we followed the more recent approach of Bansal et al. (2021) and use their word-clustering approach for creating cloze-style tasks. We thus create millions of such self-supervised tasks and combine

them with supervised GLUE tasks for training the Meta-Adapters. In an episode of training, we sample a GLUE task with probability $\lambda$ or a self-supervised task with probability $1 - \lambda$.

**Summary:.** Meta-Adapters are meta-learned adapter layers that are trained to enable parameter efficient few-shot learning. They are inserted in a pre-trained transformer and used alongside the regular adapters. The training of Meta-Adapters proceeds in meta-learning episodes. In each episode a *training task* is sampled, the adapters are fine-tuned on the task data (4) and the performance of the fine-tuned model, as evaluated by the loss on task's validation data, is used as the error to train (5) the parameters of the Meta-Adapters. In addition, this training also learns the initialization of the adapters used for fine-tuning along with the learning rate to use for fine-tuning the adapters. At inference time, parameters of the pre-trained model and the Meta-Adapters are fixed, and the adapters are fine-tuned for each *target task* using the learned learning rates.

## 4 Experiments

In this section, we evaluate the Meta-Adapters for their utility in few-shot learning of new unseen tasks and compare them with contemporary methods that utilize adapters as well as meta-learning methods for few-shot learning.

### 4.1 Experimental Setup

Unlike existing applications of adapters (see section. 5), our work evaluates the utility of adapters in a transfer learning setting where only few examples are available for each task. For this, we consider a suite of 17 downstream classification tasks. The tasks are obtained from the few-shot datasets released* by prior work on few-shot learning (Bansal et al., 2020a), making our results comparable with previously published results on these tasks. All evaluations are in the $k$-shot setting, with $k = 4, 8, 16$, where $k$ is the number of examples per label.

**Evaluation Tasks:.** The downstream classification tasks fall into the following categories: (1) Sentiment classification (4 tasks): 4 domains of sentiment classification on Amazon reviews; (2) Rating classification (5 tasks): 4 domains of ternary rating classification (high, medium, low) on Amazon reviews and classifying tweets about Airline into ternary sentiment; (3) Entity typing (2 tasks): two domains (news and restaurant queries) of classifying phrases in a sentence into entity types; (4) Natural language inference (1 task): scientific domain dataset for entailment classification; (5) Political classification (3 tasks): categorizing tweets into whether or not it has a political bias, classifying the intended audience for a political tweet (constituency, national), and classifying the substance of the text into fine-graned topics; (6) Other text classification (2 tasks): classifying tweets into whether or not they indicate a disaster and fine-grained classification into emotions.

**Models Evaluated:.** We evaluate some state-of-the-art models for both parameter-efficient fine-tuning as well as few-shot learning in our experimental setup. We consider the following models:

1. Adapter (Houlsby et al., 2019): The original adapter that only fine-tunes the adapter parameters.

2. Adapter-Fusion (Pfeiffer et al., 2021): A recent approach that trains adapters on multiple tasks, e.g. GLUE tasks, and then learns to compose them using attention mechanism (see section 5).

3. Hybrid-SMLMT (Bansal et al., 2020b): A meta-learning approach for few-shot learning that fine-tunes almost all parameters and does not include any adapters.

4. Meta-Adapters: the proposed model

---

*https://github.com/iesl/leopard

**Implementation Details:.** Note that Adapter-Fusion (Pfeiffer et al., 2021) wasn't evaluated in the few-shot setting, however, since it combines many trained multi-task adapters together, it can be a competitive alternative for few-shot scenarios. We use their released GLUE fusion adapters and their released code for evaluations. For fair comparisons, Adapter-Fusion and Hybrid-SMLMT only use GLUE supervised tasks for their training. All the compared methods use the same underlying BERT model, so that differences in performance are not due to using different models. We use the released Hybrid-SMLMT code to train this model as the released model used cased BERT model while all the other models used uncased BERT models. Our implementation results are comparable with those reported in Bansal et al. (2020b). Note that Hybrid-SMLMT fine-tunes about half of the parameters, as they found it beneficial to freeze alternate layers during fine-tuning (Bansal et al., 2020b). Hyper-parameters for the Meta-Adapters are available in the Supplementary. Code and trained models are available [†].

## 4.2 Results

We evaluate the baseline models and the proposed approach on the evaluation tasks. Each task is evaluated using 10 random few-shot training sets for $k = 4, 8, 16$, totalling 510 evaluations across the 17 tasks for each model. First, we summarize the overall results across all the tasks. Then we perform several ablations to better understand the performance of Meta-Adapters.

**Overall Results.** The overall results on all the tasks can be seen in Fig. 1. Here we analyze the overall average performance across the 17 tasks, to get an estimate of how the models compare on the two axes of few-shot accuracy and parameter efficiency. On parameter efficiency, the Meta-Adapters are orders of magnitude more efficient than both Adapter-Fusion (5%) and Hybrid-SMLMT (0.6%). Since we use a significantly smaller bottleneck size than Adapter, the Meta-Adapters are also more efficient than Adapter. We show in ablations later that Adapter perform worse when compared to similar size Meta-Adapters. This indicates that Meta-Adapters can enable increased parameter efficiency without compromising on accuracy. Now, lets look at the overall few-shot accuracy and first consider the 4-shot setting. Interestingly, not only are the Meta-Adapters most efficient, they perform just as accurately as the best performing baseline model, Hybrid-SMLMT, that does full fine-tuning. In the 8-shot setting, Meta-Adapters are still competitive with full fine-tuning, albeit slightly worse, and better than both the parameter-efficient baselines, Adapter and Adapter-Fusion, by a large margin. Note, that Adapter-Fusion are better at transfer learning than regular Adapter, however, they are less parameter-efficient than the other models.

**Results on Individual Tasks.** Table 1 shows the results on the individual tasks. For sentiment and rating classification tasks on Amazon reviews, we show the average results across the 4 domains to avoid repetition of related tasks. In the 4-shot setting, Meta-Adapters performance is better than all the other parameter-efficient methods on 9 out the 11 task types, and is competitive with the full fine-tuning approach. In the 8-shot setting, Meta-Adapters are better than Adapter or Adapter-Fusion in 7 out of the 11 task types. Overall, these results indicate that Meta-Adapters lead to accurate few-shot learning compared to other parameter-efficient alternatives. Compared to full fine-tuning, we see that Meta-Adapters perform competitively on most tasks, and the largest drop in accuracy is on the Scitail task.

**Comparison with prompting (GPT-3).** An alternative to fine-tuning for few-shot learning that has been very successful is few-shot prompting (Radford et al., 2019; Brown et al., 2020). While prompting saves on the cost of fine-tuning the model, it typically requires very large models to work competitively (Brown et al., 2020) and suffers from poor inference cost due to the larger model size. We compare fine-tuning Meta-Adapters with few-shot prompting the GPT-3 models. Since querying

---

[†]https://github.com/theTB/meta_adapters

| Task | N | k | Adapter 0.03x | Adapter-Fusion 0.41x | HSMLMT 1.00x | Meta-Adapters 0.01x |
|---|---|---|---|---|---|---|
| CoNLL | 4 | 4 | 53.4 ± 7.8 | 41.6 ± 4.4 | 59.9 ± 5.4 | 64.1 ± 2.9 |
| | | 8 | 69.2 ± 4.0 | 63.6 ± 5.8 | 70.4 ± 3.5 | 71.3 ± 3.1 |
| | | 16 | 78.1 ± 3.5 | 78.4 ± 3.8 | 79.4 ± 1.5 | 77.9 ± 1.4 |
| Restaurant | 8 | 4 | 50.0 ± 4.3 | 36.5 ± 4.3 | 56.3 ± 3.7 | 55.9 ± 5.0 |
| | | 8 | 70.6 ± 2.8 | 61.3 ± 8.6 | 70.0 ± 2.4 | 67.6 ± 2.5 |
| | | 16 | 76.6 ± 3.1 | 68.7 ± 6.2 | 76.8 ± 2.2 | 73.9 ± 1.7 |
| Airline | 3 | 4 | 51.2 ± 9.7 | 62.7 ± 6.1 | 60.6 ± 6.8 | 60.9 ± 5.8 |
| | | 8 | 61.1 ± 8.3 | 67.1 ± 4.6 | 66.9 ± 6.2 | 66.3 ± 3.1 |
| | | 16 | 68.3 ± 4.2 | 69.1 ± 3.0 | 70.1 ± 3.1 | 67.3 ± 2.6 |
| Disaster | 2 | 4 | 56.1 ± 6.4 | 56.6 ± 7.7 | 63.1 ± 8.0 | 61.6 ± 10.1 |
| | | 8 | 62.7 ± 6.5 | 60.8 ± 7.4 | 66.3 ± 4.9 | 66.1 ± 4.8 |
| | | 16 | 69.1 ± 3.0 | 65.5 ± 7.1 | 72.1 ± 3.2 | 70.7 ± 3.8 |
| Political Audience | 2 | 4 | 51.9 ± 3.1 | 51.8 ± 3.1 | 55.9 ± 4.8 | 57.0 ± 4.9 |
| | | 8 | 55.6 ± 2.7 | 57.1 ± 4.5 | 59.6 ± 4.6 | 59.9 ± 2.8 |
| | | 16 | 61.3 ± 4.5 | 57.0 ± 3.8 | 62.6 ± 3.7 | 62.7 ± 2.5 |
| Political Bias | 2 | 4 | 60.0 ± 6.0 | 56.3 ± 6.1 | 60.3 ± 7.6 | 61.2 ± 6.9 |
| | | 8 | 62.0 ± 4.8 | 61.9 ± 4.2 | 65.8 ± 4.9 | 62.7 ± 5.4 |
| | | 16 | 65.5 ± 3.3 | 65.5 ± 3.7 | 68.5 ± 2.1 | 66.4 ± 2.3 |
| Political Message | 9 | 4 | 17.6 ± 2.0 | 19.6 ± 2.2 | 17.5 ± 2.0 | 18.0 ± 1.8 |
| | | 8 | 20.7 ± 1.8 | 20.9 ± 2.7 | 19.5 ± 2.0 | 19.8 ± 2.0 |
| | | 16 | 24.2 ± 2.2 | 23.6 ± 3.2 | 21.6 ± 2.5 | 20.6 ± 1.8 |
| Emotion | 13 | 4 | 11.6 ± 1.3 | 11.7 ± 1.8 | 12.2 ± 1.3 | 12.3 ± 1.7 |
| | | 8 | 14.3 ± 1.7 | 15.6 ± 2.7 | 13.7 ± 1.6 | 12.8 ± 0.9 |
| | | 16 | 15.9 ± 1.0 | 16.4 ± 2.3 | 14.9 ± 0.9 | 13.2 ± 1.1 |
| Scitail | 2 | 4 | 53.8 ± 6.5 | 53.7 ± 05.9 | 80.0 ± 4.9 | 78.4 ± 4.3 |
| | | 8 | 58.4 ± 4.3 | 57.4 ± 10.2 | 82.0 ± 1.0 | 78.1 ± 1.8 |
| | | 16 | 64.3 ± 4.7 | 70.5 ± 4.4 | 82.8 ± 1.0 | 79.5 ± 2.2 |
| Amazon Sentiment | 2 | 4 | 60.7 ± 6.3 | 80.7 ± 2.9 | 81.7 ± 2.9 | 81.7 ± 2.7 |
| | | 8 | 66.5 ± 6.3 | 80.3 ± 4.9 | 83.9 ± 1.1 | 82.4 ± 2.1 |
| | | 16 | 75.4 ± 4.5 | 82.7 ± 2.5 | 84.3 ± 1.1 | 83.5 ± 1.0 |
| Amazon Rating | 3 | 4 | 43.5 ± 8.3 | 52.9 ± 9.7 | 56.6 ± 8.0 | 55.8 ± 7.3 |
| | | 8 | 45.2 ± 7.2 | 58.0 ± 5.9 | 59.3 ± 5.4 | 57.8 ± 5.7 |
| | | 16 | 53.7 ± 5.2 | 61.3 ± 3.1 | 62.0 ± 3.0 | 60.9 ± 3.8 |
| Overall Average | | 4 | 48.4 | 56.8 | 59.9 | 60.0 |
| | | 8 | 54.2 | 59.9 | 64.0 | 62.7 |
| | | 16 | 61.1 | 64.2 | 66.7 | 65.3 |

Table 1: $k$-shot accuracy on downstream classification tasks not seen in training. 0.01x indicates that the model fine-tunes 1% parameters per task compared to Hybrid-SMLMT.

the GPT-3 models has non-trivial costs[‡], especially for the largest model, we use a subset of the tasks for comparison and only compute accuracy on 2 of the 10 random splits for each task. We compare across all the four model sizes, all of which are orders of magnitude larger than the BERT-base model used in Meta-Adapters. Prompts used for GPT-3 evaluations are given in the supplementary. Note that the comparisons were done with the original GPT-3 model series – Ada, Babbage, Curie, DaVinci (as of December 2021) and not with the newer models available in the OpenAI API. Results are shown in Table 2. The fine-tuning based transfer-learning baselines considered here perform better than the three smaller sized GPT-3 models and are competitive with the largest model. In particular, on average, the meta-learning method HSMLMT which involves full fine-tuning performs best, while only using a pre-trained transformer that is 1590× smaller than the GPT-3 DaVinci model. Meta-Adapters retains that performance while only fine-tuning a fraction of the parameters. These results show that fine-tuning Meta-Adapters is a promising alternative to prompting giant transformer models which have significant latency and compute costs.

| Task | GPT-3 | | | | Adapter-Fusion | HSMLMT | Meta-Adapter |
| | Ada | Babbage | Curie | DaVinci | | | |
| | 350M[*] | 1300M[*] | 6700M[*] | 175000M | 110M | 110M | 110M |
|---|---|---|---|---|---|---|---|
| CoNLL | 36.9 | 39.5 | 68.4 | 80.0 | 67.8 | 71.2 | 72.0 |
| Sentiment | 75.2 | 71.5 | 90.0 | 94.3 | 83.4 | 82.7 | 84.2 |
| Airline | 52.5 | 48.2 | 68.9 | 67.1 | 69.7 | 64.9 | 68.4 |
| Political Bias | 50.3 | 50.3 | 65.8 | 62.9 | 66.9 | 67.2 | 63.9 |
| Scitail | 56.4 | 60.7 | 56.1 | 60.1 | 62.7 | 82.6 | 79.6 |
| Overall Average | 54.3 | 54.0 | 69.8 | 72.9 | 70.1 | **73.7** | **73.6** |

Table 2: Comparing fine-tuning with few-shot prompting. The GPT-3 models use few-shot prompting while the others use fine-tuning. Sizes of the models are shown below their names. On average, meta-learned models which fine-tune perform better while being much smaller in size. Meta-Adapter performs competitively with the largest GPT-3 model that is 1590× its size. [*]sizes for the smaller GPT-3 models are guess estimates from Gao (2021).

**Summary:.** Meta-Adapters are the most parameter-efficient (Figure 1), fine-tuning only 0.6% of total model parameters per task, and are more accurate at few-shot fine-tuning than competitive approaches of Adapter and Adpater-Fusion while using less parameters to fine-tune. Table 3, summarizes key properties of the various models evaluated. Meta-Adapters is also faster in training time compared to Hybrid-SMLMT, a full fine-tuning

| Model | Adapter Size | Trainable Params | Fine-tuned Params / Task | Meta-Training Speedup |
|---|---|---|---|---|
| Hybrid-SMLMT | — | 110,270,354 | 53,582,721 | 1.00x |
| Meta-Adapters | 8 | 1,453,588 | 351,936 | 0.75x |
| Meta-Adapters | 16 | 2,043,796 | 647,040 | 0.85x |
| Adapter-Fusion | 48 | 7,457,853 | 21,844,226 | — |
| Adapter | 48 | — | 1,486,658 | — |

Table 3: Summary of sizes of adapters, trainable adapter parameters, fine-tuned adapter parameters and the speedup in training when using Meta-Adapters compared with Hybrid-SMLMT.

based meta-learning approach, as Meta-Adapters have lesser number of parameters to train. We also compared Meta-Adapters with few-shot prompting of a giant pre-trained transformer GPT-3 (Table 2) and found that Meta-Adapters perform better while working with much smaller transformers.

### 4.3 Ablations

---

[‡]https://openai.com/api/pricing/

We analyze how the performance of Meta-Adapters and the baselines varies with some crucial hyper-parameters. We consider validation data from 3 tasks: CoNLL, Scitail, and Amazon Electronics, to perform the ablations and report the overall average accuracy using 10 different few-shot training sets for each task.

| Model | Vocab | Adapter Size | 4-shot | 8-shot |
|---|---|---|---|---|
| Adapter | Uncased | 48 | 55.6 | 64.3 |
| Adapter | Uncased | 16 | 55.1 | 57.6 |
| MAML-Adapters | Cased | 16 | 66.1 | 72.5 |
| Meta-Adapters | Cased | 16 | 68.2 | 74.6 |
| Meta-Adapters | Uncased | 8 | 69.7 | 74.6 |
| Meta-Adapters | Uncased | 16 | 74.6 | 77.5 |
| Meta-Adapters | Uncased | 32 | 70.3 | 76.5 |

Table 4: Ablations for Meta-Adapter.

**Meta-learning Adapter initialization without Meta-Adapters.** First we consider whether Meta-Adapters contribute to improvements in few-shot learning. For this we consider a meta-learning model that skips the Meta-Adapters altogether but still learns an initialization of adapter modules for few-shot fine-tuning. This approach is akin to adding adapter to an existing model and using the MAML (Finn et al., 2017) approach to learn their initialization. Table 4 compares Meta-Adapters with this ablation, termed MAML-Adapters. We can see that this leads to a large drop in average accuracy in both 4-shot and 8-shot settings, while there is no other benefit in parameter-efficiency from this approach. This shows that Meta-Adapters help in improving the few-shot accuracy.

**Size of Adapter and Meta-Adapters.** Next we consider how the sizes of the adapters effect accuracy. Prior work on Adapter have explored this in-depth (Houlsby et al., 2019; Pfeiffer et al., 2021), and larger adapters often work better. We consider two size of adapters, 48 and 16. We use size 48 as it is also the size that worked best for Adapter-Fusion and we use the smaller size 16 to compare with the Meta-Adapters. Note that in the few-shot setting, it is not feasible to find the best size for each given task, as in prior work (Houlsby et al., 2019), due to unavailability of validation data. Comparing the two Adapter sizes, in Table 4, we find that larger adapter performs better, specially in the 8-shot setting. However, Meta-Adapters allow comparatively better accuracy even with increased efficiency. We can see that at the same size of 16, Meta-Adapters is better by a large margin than Adapter. As we vary the size of the Meta-Adapters, we find that even at the smaller size of 8, they are still better than Adapter of size 16, 48. Interestingly, we observed better performance of Meta-Adapters at size 16 than at size 32.

**Effect of model vocabulary.** An interesting axis that affects overall performance is the choice of the pre-trained model vocabulary. We explored cased and uncased BERT-base models in conjunction with Meta-Adapters. We found that the uncased models consistently performed much better than the cased models (Table 4). This is likely because the downstream classification tasks often contain noisy user generated text. The choice of uncased BERT model also makes our results comparable with prior work (Pfeiffer et al., 2021).

## 5 Related Work

Since their introduction, adapters (Houlsby et al., 2019) have been widely applied (Houlsby et al., 2019; Stickland and Murray, 2019; Bapna and Firat, 2019; Rücklé et al., 2020) as a parameter-efficient finetuning method for large transformer-based (Vaswani et al., 2017) pre-trained models, such as BERT (Devlin et al., 2019). Prefix-tuning (Li and Liang, 2021), also known as prompt-tuning (Lester et al., 2021), is another line of popular light-weight finetuning methods which fine-tune continuous task-specific representations while keeping the large pre-trained parameters untouched. In contrast to adapters which insert task-specific parameters in between layers, these models pre-pend a trainable task-specific representations to either the input layer (Lester et al., 2021) or on every layer (Li and Liang, 2021). While these methods are promising in terms of parameter-efficient finetuning methods, with its active research progress in multi-task (Houlsby et al., 2019; Stickland and Murray,

2019) and transfer learning (Pfeiffer et al., 2020), we choose adapter framework to develop our proposed approach as prompt-tuning has been shown to only exceed fine-tuning at very large model scales (Lester et al., 2021).

Multi-task adapter (Stickland and Murray, 2019) is perhaps the first work that applied adapters to multi-task learning. In this framework, given $M$ tasks, pre-trained parameters $\theta$ are fine-tuned along with a set of $M$ task-specific parameters. However, in follow-up work, Adapter-Fusion (Pfeiffer et al., 2021) shows that a model that simply combines adapters from multiple tasks through attention, without updating the pre-trained model $\theta$, performs better than multi-task adapters. The idea in Adapter-Fusion is that rather than fine-tuning the shared $\theta$ parameters for multi-task, they instead learn an adapter-fusion layer that combines all $M$ source task adapters to benefit each of the tasks. While Adapter-Fusion has the capability to transfer to unseen target tasks outside of the $N$ source tasks, Pfeiffer et al. (2021) only test it when target task is part of the source tasks. In this paper, by choosing Adapter-Fusion as our baseline, we test its efficacy in few-shot learning of new target tasks. While Adapter-Fusion is much more efficient than multi-task adapters, it uses a larger amount of parameters compared to standard adapters due to fusion layers working on the full dimension of the pre-trained model, e.g. 768 for BERT-base.

Within meta-learning literature (Hospedales et al., 2020), our work is related to methods (Kossaifi et al., 2019; Flennerhag et al., 2020) that embed tensor projections in convolution networks for improved gradient conditioning in a meta-learning model. Other approaches (Mishra et al., 2018; Zintgraf et al., 2019; Lee and Choi, 2018; Raghu et al., 2019; Oh et al., 2020; Chen et al., 2021) have explored meta-learning with shared paramaters across tasks with goals of better convergence or avoiding over-fitting. However, these prior methods don't leverage pre-trained models and are not developed for parameter-efficient fine-tuning. In particular, Raghu et al. (2019) found that on computer vision meta-learning benchmarks only adapting the classifier head is sufficient. Subsequent work showed that this does not hold on more difficult benchmarks (Oh et al., 2020) or the NLP tasks (Bansal et al., 2020a) considered here.

Meta-learning methods (Vinyals et al., 2016; Santoro et al., 2016; Finn et al., 2017) have often been employed to enable better few-shot learning on many NLP tasks (Han et al., 2018; Gao et al., 2019; Dou et al., 2019; Bansal et al., 2020a,b; Ye et al., 2021). We compare with a recent few-shot learning work in NLP (Bansal et al., 2020b) that uses the MAML (Finn et al., 2017) approach on self-supervised tasks for few-shot classification. Their approach isn't parameter efficient whereas the proposed approach using Meta-Adapters performs comparably with a fraction of parameters for fine-tuning. Alternative methods for few-shot learning include very large pre-trained language models like GPT-3 (Brown et al., 2020) that don't fine-tune any parameters and use natural language prompts for few-shot learning. However they can be sensitive to prompt-orders (Lu et al., 2021), have a limited context length due to which they don't scale to larger datasets, and have high latency in inference due to their size. Extensions of Meta-Adapters to the soft-prompting approach (Li and Liang, 2021), in few-shot settings, can be a promising avenue for future work.

## 6  Limitations and Broader Impact

We introduced Meta-Adapter, a parameter-efficient fine-tuning method for few-shot learning that is competitive with contemporary transfer learning methods while only fine-tuning a fraction (0.6%) of the model parameters for each task. Thus, Meta-Adapter can be deployed to serve hundreds of tasks simultaneously with a shared pre-trained model, while only doubling the total number of parameters post fine-tuning. While the Meta-Adapter layers requires additional training, they make downstream fine-tuning more efficient, reducing the carbon footprint for fine-tuning which can quickly surpass the pre-training footprint when these models are served for millions of customers to fine-tune on their tasks.

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

## A   Additional Implementation Details

Hyper-parameters used in the meta-training phase are given in Table 5.

For fine-tuning on target tasks we tune need to specify the number of steps. Instead of tuning the number of steps for Meta-Adapter and Hybrid-SMLMT Bansal et al. (2020b), we found it better to instead tune a training loss threshold and fine-tune until the loss reaches that threshold. The loss thresholds for Meta-Adapter are as follows: (1) 4-shot: 1e-3 ; (2) 8-shot: 1e-2 ; (3) 16-shot: 1e-2. Following Bansal et al. (2020b), we use a batch-size of 4 and scale the batch-size with the number of labels per task.

Fine-tuning hyper-parameters for adapters and adapter-fusion include the learning rate and number of epochs. We sweep over values for learning rates in $\{1e-3, 1e-4, 1e-5\}$ and epochs in $\{10, 20, 50, 100, 150, 200\}$ to pick the best hyper-parameters for each $k$-shot.

| Hyper-parameter | Value |
|---|---|
| Tasks per batch | 16 |
| Attention dropout | 0.1 |
| Hidden Layer Dropout | 0.1 |
| Outer Loop Learning Rate | 1e-05 |
| Inner Loop Steps | 6 |
| Meta-training Steps | 540k |
| Lowercase text | True |
| Sequence Length | 128 |
| Learning-rate Warmup | 10% of steps |
| Number of SMLMT Tasks | 4 Million |
| $\left|\mathcal{D}_T^{tr}\right|$ | 60 |
| $\left|\mathcal{D}_T^{val}\right|$ | 10 |
| Number of classes for SMLMT tasks | [2,3,4,5] |
| GLUE vs SMLMT sampling ratio $\lambda$ | 0.25 |

Table 5: Hyper-parameters used in meta-training.

## B  Datasets

All dataset sources used in meta-training or evaluations are publicly available.

Evaluation dataset splits were taken from Bansal et al. (2020a). Data statistics and original sources are listed in Table 6.

Unsupervised meta-training tasks were constructed following Bansal et al. (2020b) from the Wikipedia corpus dump at: `https://www.tensorflow.org/datasets/catalog/wikipedia`. Example of a task is given in Fig. 3.

Figure 3: Example of a cloze-style task in SMLMT from Bansal et al. (2020b).

| Dataset | Labels | Training Size | Validation Size | Testing Size | Source |
|---|---|---|---|---|---|
| Amazon Review Domains | 2 | 800 | 200 | 1000 | Blitzer et al. (2007) |
| MRPC | 2 | 3669 | 409 | — | Dolan and Brockett (2005) |
| RTE | 2 | 2491 | 278 | — | Dagan et al. (2005); Haim et al. (2006); Giampiccolo et al. (2007, 2008) |
| Scitail | 2 | 23,596 | 1,304 | 2,126 | Khot et al. (2018) |
| Airline | 3 | 7320 | — | 7320 | https://www.figure-eight.com/data-for-everyone/ |
| Disaster | 2 | 4887 | — | 4887 | https://www.figure-eight.com/data-for-everyone/ |
| Political Bias | 2 | 2500 | — | 2500 | https://www.figure-eight.com/data-for-everyone/ |
| Political Audience | 2 | 2500 | — | 2500 | https://www.figure-eight.com/data-for-everyone/ |
| Political Message | 9 | 2500 | — | 2500 | https://www.figure-eight.com/data-for-everyone/ |
| Emotion | 13 | 20000 | — | 20000 | https://www.figure-eight.com/data-for-everyone/ |
| CoNLL | 4 | 23499 | 5942 | 5648 | Sang and De Meulder (2003) |
| MIT-Restaurant | 8 | 12474 | — | 2591 | Liu et al. (2013) https://groups.csail.mit.edu/sls/downloads/restaurant/ |

Table 6: Dataset statistics for all the datasets used in our analysis. "-" represent data that is either not available or not used in this study.

## C  Prompts for GPT-3

**SciTail**

Please classify a piece of text into categories.

Text: Skin Mesh Human skin has a layered structure consisting of the dermis and epidermis.
Question: Most skin structures originate in the dermis. True or False?
Answer: False
− − −

Text: The four basic tissue types are epithelial tissue, connective tissues, nervous tissue, and muscle tissue.
Question: Four types of tissue are found in animals. True or False?
Answer: True
− − −

Text: Trees produce oxygen as a byproduct through the photosynthesis process.
Question: Oxygen is made by trees and other plants during photosynthesis. True or False?
Answer:

---

**Sentiment Electronics**

Please classify a piece of text into categories.

Text: The mouse is perfect for games. I use it to play ET and is great. The software provided by Logitech is configurable in all ways
Category: Positive
− − −

Text: I purchased this product and couldn't get it to work on a PC, laptop, or pda ( cingular 8125 ). Not only does it not register in any SD / MiniSD card reader, but the craftsmanship appears to be suspect as well. I am currently pursuing a refund
Category: Negative
− − −

Text: Code length of this product is very small. I had to buy an extention cord. The sound quality is not bad
Category:

---

**Airline**

Please classify a piece of text into categories.

Text: @SouthwestAir yall still fly in the cold right?
Category: Neutral

---

Text: @SouthwestAir Great, thank you. Best of luck dealing with this horrible winter.
Category: Positive
---

Text: @united I was on UA3782 and it was Cancelled Flightled. I'm waiting at customer service.
Category: Negative
---

Text: @SouthwestAir please reply to my DM
Category:

---

**Political Bias**
Please classify a piece of text into categories.

Text: The 1st Amendment protects # ReligiousFreedom for everyone & amp ; no American should be compelled to violate their convictions. #HobbyLobby
Category: Political
---

Text: Thanks for your support! MT  SenOrrinHatch : Today is # NationalPediatricBrainCancer-AwarenessDay. Hope you'll join me in fighting this disease
Category: Neutral
---

Text: Regrettably, the House failed to approve its proposal for a new #FarmBill. Frankly, I was shocked by the outcome.
Category:

---

**CoNLL Entity Typing**
Please classify a piece of text into categories.

Text: India. Classify: India.
Category: Location
---

Text: Leading rider Jason Weaver received a 21 - day ban from the disciplinary committee of the Jockey Club on Wednesday. Classify: Jason Weaver.
Category: Person
---

Text: Balloting inside Bosnia is scheduled for September 14, when citizens are slated to elect municipal and cantonal assemblies, separate Moslem - Croat and Serb parliaments, a national House of Representatives and a three - man Presidency. Classify: House of Representatives.
Category: Organization
---

Text: Leading stories in the Greek financial press :. Classify: Greek.
Category: Other
---

Text: HELIBOR INTEREST RATES LARGELY UNCHANGED. Classify: HELIBOR.
Category:

Table 7: Prompts used for GPT-3 on the different tasks.

