# OpenReview forum: "Meta-Adapters: Parameter Efficient Few-shot Fine-tuning through Meta-Learning"
_automl.cc/AutoML/2022/Track/Main — AutoML-Conf 2022 (Main Track)_

### Official Review · Reviewer_Heid · 2022-04-01

**Potential Impact On The Field Of Automl Rating:** 3
**Technical Quality And Correctness:** The technical quality of the presente…
**Technical Quality And Correctness Rating:** 4
**Clarity:** The paper is presented in a clear way…
**Clarity Rating:** 4

**Summary Of Contributions:**

The paper proposes the usage of meta-learning for parameter efficient fine-tuning using adapters within a pretrained transformer model. To enable the usage of adapters in the few-shot setting, additional meta-adapters are introduced that trained during meta training alongside with the adapter weights and the fine-tune learning rate. During meta testing, the meta-adapters are frozen as well and only the adapter weights are fine-tuned from the meta-learned initialization. A large evaluation on many different NLP tasks shows superior performance to other parameter efficient fine-tuning methods and even comparable performance to full fine-tuning, in particular for very low sample sizes.

**Overall Review:**

The proposed approach to combine meta learning with the parameter-efficient adapter method for fine-tuning that allows to use a single frozen backbone with task specific adapters is novel to my knowledge and provides an interesting extension of simply meta-training the adapters. The proposed approach is nicely presented and easy to follow. Furthermore, the paper provides a large experimental evaluation on many different NLP tasks and comparison with different baseline methods coming that have been proposed in the fine-tuning and meta-learning context. The following are my additional questions and recommendations:

1. How effective is meta-learning the fine-tuning learning? An ablation study would be interesting (e.g., what are the meta-learned learning rates vs. the learning rates found by grid search for the adapter-only networks).
2. What is the importance of meta-adapter placement around the regular adapter layers? Would it suffice to have a single meta-adapter either before or after the regular adapter?
3. For the ablation study that does meta-learning without meta-adapter (i.e., the MAML-Adapter case) it would be interesting to see the MAML-Adapter performance for uncased data and multiple adapter sizes as well as there seems the performance seems to be quite sensitive to the adapter size.
4. Table 2 and 3: The usage of meta-adapter in conjunction with adapter size in this table can be confusing: Does the adapter size relate to the bottleneck size of the meta-adapter blocks, the adapter blocks, or both in a meta-adapter network? If it refers only to the size of the meta-adapter blocks, how large are the adapter blocks in a meta-adapter network and what is the reason for the difference in fine-tuned params for meta-adapter and adapter networks in table 3?
5. What would be the effect of fine-tuning the meta-adapter block parameters as well? Could this lead to further improvements in settings for large k (e.g., k=16)?

Minor:

1. Line 223: How do the 340 evaluations arise given there 10 training sets, 3 possible values for k and 17 tasks?


**Potential Impact On The Field Of Automl:**

The paper nicely combines meta-learning with parameter efficient fine-tuning and has good experimental results.

**Reproducibility:**

The paper provides a reproducibility list that is filled out in a reasonable way and will provide models and code to reproduce the experiments.

**Review Confidence:**

3: You are fairly confident in your assessment. It is possible that you did not understand some parts of the submission or that you are unfamiliar with some pieces of related work.

**Review Rating:**

5: Accept, good paper

**Review Summary:**

The paper presents in interesting idea combining meta-learning with a parameter efficient method for fine-tuning. It provides extensive experimental evaluation of the effectiveness of the approach.

---

### Official Review · Reviewer_r3Zo · 2022-04-03

**Potential Impact On The Field Of Automl Rating:** 2
**Technical Quality And Correctness Rating:** 2
**Clarity Rating:** 3

**Summary Of Contributions:**

+ this work generalizes the transfer-learning technique for pretrained models Adapter (Houlsby et al., 2019) to the meta-setting, by adding Meta-Adapters. In few-shot learning settings, these Meta-Adapters, which are meta-learned but not fine-tuned, serve the purpose of easing optimization.
+ competitive performance with state-of-the-art (sota) few-shot learning methods on NLP benchmarks
+ performant on NLP benchmarks while only requiring to fine-tune 0.6% of the parameters compared to a sota approach.

**Clarity:**

\- background section and related work should be merged
\- Writing can be improved:
  \(1) the way methods are used in plural or singular form is confusing. Usually, a technique is referred to in singular form, e.g., "MAML is", "Dropout has", ... . Please do the same with the technique(s) you propose or reference, or use adapted wording. E.g., "Meta-adapter fine-tune[s]", "Meta-adapter performs" and
"We introduce Meta-Adapter, which are small blocks" --> "We introduce Meta-Adapter, which [is an approach] that introduces small blocks" OR call your technique "Meta-AdapterS"/ "Meta-adapter layers" (as is done in 350) so you then you can use the plural form. One or the other.
  \(2) Also, "re-purpose existing pretrained model" --> it is not clear whether you are talking about "A model" (with missing article) or multiple "models" (with missing plural form).
  \(3) unnecessary hyphen insertion: "re-purpose", "few-steps", "hyper-parameter", ...
  \(4) "these nested minimization"
\- a major benefit of the paper is mostly hidden in the broader impact section: storage/carbon footprint reduction due to only requiring fine-tuning significantly fewer parameters. It would be great to expand further on this qualitatively and quantitatively.
\- the "Limitations ..." section discusses no limitations?

**Overall Review:**

\+ Meta-Adapter significantly reduces the number of parameters to be finetuned on top of a pre-trained model, while keeping performance mostly on par with SOTA.

major:
\- The largest drawback is that the approach is only validated on natural language processing (NLP) benchmarks, leaving the question whether the same parameter-performance characteristic can be maintained on computer vision, regression or reinforcement learning benchmarks.

\- the downstream benefit of (meta-)training fewer parameters should be discussed more. For example compare the inference time training speedup (not the meta-training speedup, which seems to be marginal compared to the parameter reduction).

minor:
\- no comparison with meta-baseline [3], a very similar technique where meta-parameters \omega are the same as \theta (in fact, a further trained version, also from a pretrained \theta).

\- no mention or comparison to ANIL/BOIL, which are very similar to Adapter and (parameter) improvements over MAML, but don't leverage pretrained models.

[3] https://arxiv.org/abs/2003.04390

**Potential Impact On The Field Of Automl:**

\+ Demonstrated that generalizing the idea of Adapter (Houlsby et al., 2019) to the meta-setting does not deteriorate performance by much (compared to (H)SMLMT).
\+ Although performance is mostly on-par or slightly below sota, the fact that only a small fraction of the parameters needs to be fine-tuned (0.6% in paper) could be beneficial in large-scale AutoML systems.

\- However, Table 3 (speedup) shows that the meta-training speedup compared to (H)SMLMT is surprisingly low: still 75% training time for only 0.6% of the parameters. Thus, beyond storing less task-specific parameters and an unquantified possible carbon footprint reduction in large-scale systems, the approach does not seem to have much motivated substantial benefit. A suggestion here would be to quantify and compare the test time (few-shot fine-tuning) speedup.

**Reproducibility:**

The reproducibility list is filled out in a reasonable way.

I believe that I and other researchers would be able to reproduce the results.

**Review Confidence:**

3: You are fairly confident in your assessment. It is possible that you did not understand some parts of the submission or that you are unfamiliar with some pieces of related work.

**Review Rating:**

2: Reject, not good enough

**Review Summary:**

Meta-adapter is an extension of Adapter (Houlsby et al. 2019) to the meta-learning setting. Similar to Adapter, Meta-adapter enables to fine-tune a significantly lower number of parameters for task adaptation by introducing intermediate layer modules. However, it does so while maintaining performance in the few-shot setting, unlike Adapter, as shown in this paper. The main drawback of the work is that it is only evaluated on NLP tasks. To make general few-shot AutoML conclusions, the approach needs to be evaluated at least also on one (more) common meta-learning computer vision benchmark ((mini/tiered)-ImageNet, ...). Next to this, the writing/clarity should also be fixed, and the benefit of fewer parameters for few-shot learning better motivated. These things would make it possible to increase the score.

**Technical Quality And Correctness:**

\+ The proposed approach, theory, and experiments are sound.

\- Generalizing from only NLP benchmarks to all of meta-learning is too much of a shortcut without doing multi-domain (computer vision, ...) experiments.
\- The background section forgets to mention very relevant meta-learning methods like ANIL [1] or BOIL [2], which are techniques that also only fine-tune part of the model (i.e. not all parameters) at "test time".
\- 265: "large drop", without confidence intervals it is too quick to jump to such a conclusion; 68.2 vs 66.1 (with e.g. +- 5%) is not significant.


[1] https://arxiv.org/abs/1909.09157
[2] https://arxiv.org/abs/2008.08882

---

### Official Review · Reviewer_vR3z · 2022-04-05

**Potential Impact On The Field Of Automl Rating:** 2
**Technical Quality And Correctness Rating:** 3
**Clarity:** The paper is well written and very ea…
**Clarity Rating:** 3

**Summary Of Contributions:**

The motivation of the paper:
1. pre-trained model is very useful, but fine tuning it with all parameters is impractical.
2. Adapter method is proposed to reduce the number of model parameters to be fine tuned by adding additional task-specific parameters and keeping frozeing pretrained model. However, this method is not useful for few shot learning tasks.

Contributions:

1. This paper proposed a new parameter efficient fine tuning approach based on pretrained model based on the Adapter method on the top of meta learning idea, in particular MAML, called: Meta-Adapter.
2. They add some new blocks to the pretrained model: meta-adapter block (\w), in addition to the original pretrained model layers (\theta), and adapter (\phi).
   1)  \phi will be meta-learned, \w will be learned along with \phi, \theta will be fixed during training and inference.
   2) During inference, learned \w will be fixed, \phi will be fine tuned based on new tasks.
3. Experiments validate the effectiveness and efficiency of the proposed model.

**Overall Review:**

Positive:

1. This paper proposed a new parameter efficient fine tuning approach based on pretrained model based on the Adapter method on the top of meta learning idea, in particular MAML, called: Meta-Adapter.
2. The formulation is clear for autoML community.
3. Experiments validate the effectiveness and efficiency of the proposed model.

Negative:

Some points are not clear and should be clarified:
1. the similarities and difference between Meta-Apdater and MAML-Adapter.
    It seems that the only difference is
    - Meta-Adapter adds some new blocks to the pretrained model.
    - MAML-Adaper does not introduce new blocks.

   The similarity is to meta-learn the adapter parameters.

2. What is the specific setting for "transfer learning" in the experiment? what is source and target? These settings are important for readers.

3. Since the Meta-Adapter approach brings more parameters (meta-adapter blocks) to Adapter approach, why does Meta-Adapter have less parameters than Adapter (as shown in Fig.1)? I am confused with this.

**Potential Impact On The Field Of Automl:**

I might cite this paper if the work is very related to parameter efficiency fine tuning for few shot learning case.

**Reproducibility:**

The paper attaches supplementary including code.

I believe that I and other researchers would be able to reproduce the results.

**Review Confidence:**

3: You are fairly confident in your assessment. It is possible that you did not understand some parts of the submission or that you are unfamiliar with some pieces of related work.

**Review Rating:**

4: Marginally above the acceptance threshold (use sparsely)

**Review Summary:**

Pre-trained model is very useful, but fine tuning it with all parameters is impractical. Adapter method is proposed to reduce the number of model parameters to be fine tuned by adding additional task-specific parameters and keeping frozeing pretrained model. However, this method is not useful for few shot learning tasks.

Considering the limitation above, this paper proposed a new parameter efficient fine tuning approach based on pretrained model based on the Adapter method on the top of meta learning idea, in particular MAML, called: Meta-Adapter.

I made decision based on the positive and negative points listed above.

**Technical Quality And Correctness:**

Generally speaking, the approach seems correct. It is nothing but the bi-level optimization, although only adapter parameters are meta-learned.

1. This paper adds some new blocks to the pretrained model:
- meta-adapter block (\w)

in addition to the original
- pretrained model layers (\theta)
- adapter (\phi).

   (1)  \phi will be meta-learned, \w and learning rates will be learned along with \phi

   (2)  \theta will be fixed during training and inference.

2. Experiments validate the effectiveness and efficiency of the proposed model.

---

### Official Review · Reviewer_Jonq · 2022-04-05

**Potential Impact On The Field Of Automl Rating:** 2
**Technical Quality And Correctness Rating:** 2
**Clarity Rating:** 4

**Summary Of Contributions:**

This paper propose to meta-learn adapters as a way to efficiently finetune/adapt large pretrained (language) models to tasks at hand. In particular, the authors follow the standard adapter settings as proposed in Houlsby et al. 2019, but encase a base adapter layer with Meta-Adapters - two-layer bottleneck MLPs - which are trained during the outer loop of an otherwise standard gradient-based MAML-like Meta-Learning optimization process. Through a set of ablations, the authors are able to show that when compared to the standard non-meta-learned adapter setting their meta-adapter approach is more sample-efficient, performing much in various few-shot classification settings. A large experimental evaluation on 17 downstream few-shot classification tasks against the adapter baseline, another adapter-reference and one meta-learning baseline shows Meta-Adapter comparing favourable when accounting for the number of parameters tuned.

**Clarity:**

The paper is well written and very easy to follow. However, some form of conclusion of final summary would have been nice, although not in any way crucial.


**Overall Review:**

Positive aspects include:
* the idea of Meta-Learning through Meta-Adapters is, to the best of my knowledge, novel.
* within the adapter-domain of Meta-Learning, the proposed approach appears to perform quite favourably.
* The paper is well written and easy to follow.

For negative aspects, see concerns listed in Section 3 on the Technical Quality and Correctness.

**Potential Impact On The Field Of Automl:**

This method will have primary impact in the subfield of Meta-Learning, or more precisely memory-efficient adaptation to a collection of few-shot tasks that require the retention of each few-shot model and have significant compute constraints.
While I'm not entirely sure of it's immediate practical relevance, the paper does showcase a variation of gradient-based Meta-Learning through the re-use of pretrained models which could be of interest to researchers in this field.
I do however believe that the overall impact is limited, as other similar methods with the same idea in mind (avoid finetuning the whole model) have already been developed both utilising adapter-like settings (e.g. Li et al., "Cross-domain Few-shot Learning with Task-specific Adapters"), meta-learning with hypernetworks (e.g. Zhao et al., "Meta-Learning via Hypernetworks") or, more recently, using prompts (see e.g. Zhao et al. "Calibrate Before Use: Improving Few-Shot Performance of Language Models" or Logan et al., "Cutting Down on Prompts and Parameters: Simple Few-Shot Learning with Language Models") which tend to be even more parameter efficient. Given the lack of comparison to these references, researchers may opt for one of these other approaches.


**Reproducibility:**

Hyperparameters are provided in the supplementary, and explanations in the text offering sufficient insights to reproduce the results are given.

**Review Confidence:**

4: You are confident in your assessment, but not absolutely certain. It is unlikely, but not impossible, that you did not understand some parts of the submission or that you are unfamiliar with some pieces of related work.

**Review Rating:**

4: Marginally above the acceptance threshold (use sparsely)

**Review Summary:**

I am going with a rejection for now. While the method performs better than the adapter baselines, the overall approach is quite incremental, and it is unclear how it compares to and ties in with other similar approaches to this problem. This connects to the main issue, which is the quality of the experimental evaluation that lacks a more thorough comparison against other meaningful baselines.



**Technical Quality And Correctness:**

The proposed Meta-Adapter approach on its own is, to the best of my knowledge, novel.
While it does appear somewhat incremental, it does provide a generally sound parameter-efficient gradient-based meta-learning approach. As no major novel theoretical work was introduced, my main concerns are with the conceptual motivation and experimental evaluations.

In particular, the comparison against other Meta-Learning methods is lacking. While a large selection of benchmark comparisons is performed, only a handful of methods are compared to. While I agree that the paper primarily aims to extend adapter-style Meta-Learning, I believe that comparison to other baselines such as (1) training a linear regressor on the pretrained backbone or adapted backbone similar to e.g. Chen et al. "A New Meta-Baseline for Few-Shot Learning" or Tian et al. "Rethinking Few-Shot Image Classification", (2) prompt-based approaches as another low-memory alternative or (3) other generally known few-shot classification approach s.a. e.g. MetaOptNet.

In addition to that, a MAML-like training objective is very slow - even if only a handful of parameters are trained in each layer, full backpropagation still needs to happen. As such, it would have been great if the authors gave insights into the Meta-Learning training times and how these compared to other methods. Similarly, the percentage of finetuned parameters listed in Table 1 refers only to the parameters used for each specific method and as such don't allow for direct comparison. Further informative metrics such as absolute parameter count, walltime, flops or memory usage would give much more insights here.

---

### Meta-Review · Area_Chair_PLYy · 2022-05-08

**Recommendation:** Accept
**Confidence:** 4

**Metareview:**

I would like to thank the reviewers for constructively engaging with the submission!

`All reviewers` agree that there is value in approaches which efficiently repurpose pre-trained models to serve ever wider distributions of tasks, even when the basic principles underlying the approach are well known. While model "adapters" have long been used in transfer learning research, e.g. vision, NLP and RL, and meta-learning has been applied to a myriad of few-shot settings, it is also well understood that fast generalisation settings introduce additional considerations; for example, meta-learning with heterogeneous task distributions is known to be challenging, and tuning adapter-based methods to heterogeneous distributions of tasks particularly difficult, e.g. due to different capacity requirements for various tasks. Hence, incremental work is valuable and needed to understand such trade-offs in detail. I believe this is the conclusion of all but `Reviewer r3Zo`. I will revisit their arguments below.

In terms of impact of work on AutoML, I do agree with `all reviewers` that it may be limited or medium, but I would like to emphasize the value of the methodology followed in the paper, largely borrowed from meta-learning, for future AutoML works: a meta-objective is defined using a heterogeneous set of problems, including a left-out set of tasks which is used for final evaluations, and a difficult optimisation problem is solved in order to find appropriate learning rates and adaptation sub-spaces which are conducive to generalisation on a broad number of related problems.

I really appreciated the work that `Reviewer r3Zo` put into finding ways to improve the paper, which I strongly advise authors to implement! However, I do not agree that demonstrating applicability of their approach to other domains is mandatory for acceptance; this has actually been done, e.g. Flennerhag et al. 2020, Rusu et al. 2018 in RL and few-shot visual processing respectively.

As for my own recommendations for the authors, I suggest using the work of Baik et al. 2020 to further improve meta-learned hyper-parameters, such as learning rates. I believe that showing the value of meta-adapters vs. meta-learned hyper-parameters could improve the paper further, and would also be more in line with similar attempts in AutoML.

On balance, I would like to recommend acceptance, considering the improvements made by authors as per suggestions of reviewers.


### References:
* Flennerhag, Sebastian, Andrei A. Rusu, Razvan Pascanu, Francesco Visin, Hujun Yin, and Raia Hadsell. "Meta-Learning with Warped Gradient Descent." In International Conference on Learning Representations. 2019.
* Rusu, Andrei A., Dushyant Rao, Jakub Sygnowski, Oriol Vinyals, Razvan Pascanu, Simon Osindero, and Raia Hadsell. "Meta-Learning with Latent Embedding Optimization." In International Conference on Learning Representations. 2018.
* Baik, Sungyong, Myungsub Choi, Janghoon Choi, Heewon Kim, and Kyoung Mu Lee. "Meta-learning with adaptive hyperparameters." Advances in Neural Information Processing Systems 33 (2020): 20755-20765.

---

### Decision · Program_Chairs · 2022-05-13

Accept